# The Effects of Fruit-Derived Polyphenols on Cognition and Lung Function in Healthy Adults: A Systematic Review and Meta-Analysis

**DOI:** 10.3390/nu13124273

**Published:** 2021-11-27

**Authors:** Lillian Morton, Andrea J. Braakhuis

**Affiliations:** Faculty of Medical & Health Science, Grafton Campus, The University of Auckland, Auckland 1142, New Zealand; a.braakhuis@auckland.ac.nz

**Keywords:** polyphenols, cognition, lung function

## Abstract

Polyphenols are plant derived nutrients that influence oxidative stress and inflammation and therefore may have positive benefits on cognition and lung function. This systematic review and meta-analysis aimed to evaluate the effects of fruit derived polyphenol intakes on cognition and lung capacity in healthy adults. In August 2020 and October 2021, Medline and Google Scholar were used to search for relevant studies examining the effects of fruit derived polyphenol intakes on cognition and/or lung function in healthy adults (<70 years old). Fourteen studies related to cognition (409 healthy subjects) and seven lung/respiratory studies (20,788 subjects) were used for the systematic review using the preferred reporting items for systematic reviews and meta-analyses (PRISMA) guidelines. The meta-analysis (using six cognition and three lung function studies) indicated a protective effect on lung function from dietary intakes of fruit-derived polyphenols. Neither a benefit nor decrement from fruit-derived polyphenol intakes were detected for cognition. Human intervention trials examining the effects of polyphenol supplementation on lung function in healthy adults are scarce and intervention studies are warranted. More conclusive results are needed to provide recommendations for polyphenol supplementation to support aspects of cognition.

## 1. Introduction

The adverse effects of air pollution on health outcomes have been extensively documented and associations have been found between particulate matter and mortality and morbidity [1,2], the majority of which have been attributed to cardiovascular causes. More recently, adverse effects of ozone exposure have been described in the central nervous system (CNS) and associated with neurological disorders, such as Alzheimer’s and Parkinson’s disease [3,4], cognitive decline and dementia [5], and neuroinflammation [6]. The pathways underlying these effects are complex and poorly understood; however, particulate-induced oxidative stress repeatedly emerges as a potential mechanism in all these detrimental cardiovascular, respiratory, and cognitive actions.

The primary role of the respiratory system is to provide oxygen to body tissue, and is susceptible to oxidative stress due to the large surface area of the lungs and the environment at the alveoli [7]. Evidence for increased oxidative stress in lung inflammation and the formation of reactive oxygen and nitrogen species (RONS) within the airway [6,8] is emerging [9]. This oxidative injury remains localised to the respiratory system if the antioxidant capacity in the lungs can buffer them, but if overwhelmed, the RONS and extrapulmonary pro-inflammatory cytokines diffuse into the bloodstream and access the CNS by crossing the blood brain barrier (BBB) [3,6,10]. Respiratory dysfunction has been implicated and correlated with cognitive dysfunction, and pulmonary function and respiratory disease have been identified as risk markers for dementia [11,12,13]. Even in healthy volunteers, the acute effects of hypoxaemia results in minor deficits in complex reasoning, reaction times, and a reduced ’learning’ effect during repeated testing [12]. 

Mounting evidence has indicated that individuals who consume a diet containing high amounts of fruits and vegetables have lower risks of non-communicable disease and disease targets across pre-clinical, human intervention and epidemiological research [14,15,16], and a reduced incidence of age-associated diseases such as cardiovascular or neurodegenerative diseases [17]. This is attributed in part to the bioactive antioxidants and polyphenols they contain [14].

Polyphenols are secondary metabolites of plants and possess antioxidant activities that contribute to reducing oxidative stress and down-regulate the inflammatory response [18]. Polyphenols are categorised into four subclasses according to their phenol ring structure and number, and ring-binding elements: flavonoids, phenolic acids, lignans and stilbenes. The greatest proportion of polyphenols are flavonoids (60%) which produce further subclasses upon oxidation: catechins, flavanones, flavones, isoflavones, flavonols and anthocyanins [19]. 

Anthocyanins (responsible for the red, purple, and blue colours of plants and fruits), have physiological effects far beyond their antioxidant properties and influence an array of cell-signalling and gene expression pathways that act to reduce oxidative stress, improve neuron function, or alter the inflammatory environment [15]. Dietary intakes of polyphenols and anthocyanins have been associated with neuroprotective effects and age-related cognitive maintenance [15,20], improved cerebral haemodynamics and cerebral blood flow and enhanced aspects of cognition in older adults [20]. Large-scale epidemiological studies [21,22,23,24,25], and acute interventions using mice models [26,27,28], have also found that intakes of anthocyanin rich polyphenols are related to improved lung function, and lower the prevalence of respiratory symptoms [28]. 

While dietary polyphenol interventions have been used in studies, the target groups are typically older adults suffering from chronic disease; namely, lung disorders (COPD or asthma) and age-related cognitive decline, Alzheimer’s or Parkinson’s. More recently the effects of polyphenol interventions on younger adults have been investigated [29,30,31,32,33], but the extent to which polyphenols may provide a protective effect on cognition and lung function in young, healthy adults remains unclear. A significant gap exists in the understanding of the potential role of polyphenols in healthy adults to augment cognition and lung function.

Polyphenols derived from fruit and vegetables may mitigate some of the negative consequences of lung inflammation and its subsequent effect on cognition. While animal and epidemiological investigations appear to show some promise in fruit-derived polyphenols reducing the incidence of chronic disease, the current review is targeted to cognitive and lung function outcomes in otherwise healthy individuals. The present study aimed to systematically review the literature and conduct a meta-analysis of both lung and cognition trials investigating the effects of fruit derived polyphenol intake as fruit extracts, products, or whole food interventions on lung function and/or cognition in healthy adults.

## 2. Materials and Methods

The protocol for this systematic review and meta-analyses followed the preferred reporting items for systematic reviews and meta-analyses (PRISMA) guidelines [34].

### 2.1. Eligibility Criteria

Eligible studies needed to meet the following inclusion criteria: primary research published in English in peer-reviewed journals before August 2020, research conducted in healthy human adults aged over 18 years, use of fruit-derived polyphenol intervention and included cognitive function testing. Reviews, conference abstracts, observational studies, and papers where no full text were available were excluded, as were papers that did not report relevant outcome measures. The primary outcomes of interest included cognitive function, including measures of executive function and attention using validated assessment tools. 

Eligible lung studies were required to meet the following inclusion criteria: pre-clinical animal studies or human subjects, no chronic respiratory or other disease, only acute lung inflammation/injury, fruit-derived polyphenol intervention. Studies needed to include the following measures of lung function: forced vital capacity (FVC) and forced expiratory volume (FEV_1_). 

Reports with insufficient variance data were excluded from meta-analysis (SD, confidence interval, sample size). Appropriate full-text papers that adhered to the inclusion criteria were included in the SR. A summary of the selection process is outlined in Figure 1.

### 2.2. Data Sources and Search Strategy

A comprehensive, systematic search of Medline (Ovid) and Google Scholar was performed from inception to August 2020. The original search terms were again used to conduct a search in October 2021 to find articles of interest published in the period between August 2020 and October 2021. Medline (Ovid) incorporated MeSH terms when appropriate and google scholar terms adapted accordingly (Refer to Appendix A: Cognition—NIH Quality Assessment Tool for Controlled Trials). Patents and citations were excluded, the search limited to English language, and only the first 15 pages of returned results examined (Google Scholar). Titles and abstracts were systematically searched to identify peer-reviewed trials to identify those that indicated possible inclusion criteria, and duplicates and inappropriate texts were excluded. The full text of remaining papers was reviewed, and eligible studies included for the systematic review. Reference lists and reviews were manually searched for additional relevant studies for possible inclusion.

### 2.3. Data Extraction

Two authors (LM and AB) independently extracted data from included studies into a standardised form and cross-checked for consistency. Data extraction included the primary author’s surname, year of publication, sample size, sample description (age and gender), study duration and time points of data collection, intervention (including anthocyanin/polyphenol dose if reported), all cognitive measures performed (and sequence of testing) and study outcomes. A summary of studies and statistically significant outcomes in cognitive function are quantitatively expressed in Table 1. Data for analysis were classified and summarised by the affected cognitive domain; executive function and attention. Attention outcome measures were obtained from rapid visual information processing (RVIP) and serial subtraction tasks (three and sevens–SSTT, SSTS). The Stroop test data was used for executive function. Total scores achieved at 2 h post supplementation was used for the meta-analysis as this was the most consistent measurement period across trials. For crossover trials post-only outcomes are reported for interventions. Mean, variance data and included *p* values are reported as stated in the manuscripts. 

For the lung function studies, only the human cross-sectional studies were meta-analysed. The most highly adjusted values for lung function were extracted. Due to data reporting (and an absence of variance data for quintile one), quintile two was used as the lowest value for data extraction relative to quintile five, or lowest available data vs. highest. Difference in means ± SD/SEM (95% CI), and adjusted Odds Ratios (OR) are reported. 

### 2.4. Quality and Risk of Bias Assessment

Two reviewers (LM and AB) independently assessed the quality of included studies using the National Institutes of Health (NIH) Quality Assessment tool for Observational Cohort and Cross-Sectional Studies (NHLBI, 2019) and intervention trials, and if ratings differed then reviewers discussed the article to reach consensus. The Quality Assessment looks at the study aims, participants, study drop-outs, recruitment, inclusion/exclusion criteria, study design, outcome measures and blinding Appendix A.

### 2.5. Statistical Analysis

Review Manager (version 5.4.1, Cochrane, London, UK) calculator was used to derive the standard error for the MA, using reported variance data from the manuscript/report. Given the difficulty in comparing cognitive task data and measurement techniques, cognitive tasks were subdivided into the cognitive domains effected and separate random effects models were used to examine the effect of polyphenols on attention and executive function. Due to the heterogeneity of cognitive tasks include all effect sizes were calculated using standardised mean differences (SMD). Effect sizes (ES) were calculated by using the values at the 2-h time point as this was the most consistent measurement across studies. The calculated standard error for each manuscript/report was used to generate the standardised mean difference. To interpret the magnitude of change in the cognitive tests, and its 95% confidence interval (CI), the effect size of 0.2 is deemed small, 0.5 moderate and 0.8 large [47]. A positive ES in executive function and attention would indicate fruit-derived polyphenol supplementation improved cognitive outcomes. Assessment of heterogeneity between included studies was evaluated by the Higgins score (I^2^). Values of I^2^ were interpreted using the guidelines outlined in the Cochrane Handbook for Systematic Reviews of Interventions [48], where 0 to 40% might not be important; 30 to 60% may represent moderate heterogeneity; 50 to 90% may represent substantial heterogeneity, and 75 to 100% may represent considerable heterogeneity. The fixed effects model was adopted when the Higgins score was below 60%; otherwise, the random-effects model was used. Statistical significance was set at *p* < 0.05. In the forest plots, the squares denote individual study effects, and the large diamond the overall, or summary effect.

The extracted lung function data included the adjusted odds ratios (OR) and 95% confidence intervals (CI) which were converted to log of OR (LogOR) for the purposes of meta-analysis. Manuscripts or reports with various polyphenol types or subclasses were included in the meta-analysis as independent studies, with adjustments made to the weighting by reducing the sample size relative to the number of subclasses. 

## 3. Results

### 3.1. Study Selection

The predefined search terms, and the final combined search term in Medline returned 2039 results from the initial search. After reviewing the study titles, 251 were selected for further analysis. Following screening of abstracts for inclusion criteria, 25 articles for cognition and 38 for lung function were considered for full reading. Following the review of the full-articles, and excluding reviews, 23 papers were excluded for cognition, and 28 excluded for lung function, leaving a provisional list of two articles for cognition, and 11 for lung function (seven human, four mice). The second search (October 2021) returned 148 results, and after reviewing titles nine were selected for further analysis. All nine were excluded (four studies were reviews or special articles, four used cells or rodents and the one human trial was observational and did not use measures of cognition). The Google Scholar search provided 78 titles related to polyphenols and cognition. After reading abstracts and applying the inclusion criteria, 59 articles were excluded, and a further 15 excluded after reading full-text articles. Only five papers related to lung function were found after screening titles and abstracts, and all five were excluded after applying the inclusion criteria and reading the full text. Manual searching of full-text articles and reference lists identified an additional nine papers for inclusion in the SR. Therefore, 14 studies examining the effects of fruit-derived polyphenols on cognition were included for analysis and review, and seven human studies for lung function (Figure 1). Summaries of cognition studies are presented in Table 1 and lung studies in Table 2. 

### 3.2. Study Design and Supplementation

#### 3.2.1. Cognition

Fourteen papers examined the effect of fruit-derived polyphenol supplementation on cognition in healthy adults [29,31,32,35,36,37,38,39,40,41,42,44,45,46,51,52,53]. Studies utilised either chronic supplementation,

which we define as anything greater than 5 days [43,44], or acute supplementation [29,35,36,37,38,40,41,42,45,46,51,52], and one study was an acute-on-chronic [38] study design. Where a chronic dosing strategy included an acute dose within 2 h of cognitive function testing these were categorised as acute and only the acute data reported [37,41,44]. All studies were randomised, and the majority utilised a double-blind, placebo-controlled experimental design, with a washout period of either 5 weeks [35], 4 weeks [46], 10 days [41] or 7 days [37,51]. One study was a single-blind, parallel group experimental design [36], and another utilised a double-blind parallel group design [44]. Both parallel group studies had an experimental arm and a placebo arm.

All studies used cognitive testing batteries and involved supplementation with fruit-derived polyphenols; ten were berry-based interventions (blackcurrant or blueberry) [36,37,39,40,41,42,44,51], one Montmorency tart cherry [45], one grape seed extract [36,38], one Queen garnet plum [46], and one a polyphenol rich orange juice intervention [29]. The polyphenol content across studies varied. In studies with chronic supplementation daily for several weeks, the lowest dose of anthocyanin was 7.4 mg consumed daily for 8 weeks [46], while the highest dose was 400 mg of total polyphenols from grape seed extract taken in capsule form daily for a period of 12 weeks [38]. Studies with acute supplementation protocols typically used higher polyphenol doses ranging from 253 mg [40] to 414.2 mg [38] anthocyanins, and 143 mg [36] to 1234 mg [35] total polyphenol content. 

There were a wide range of cognitive tests utilised across studies, but all employed validated cognitive tests to determine the effects of fruit-derived polyphenol intervention on cognitive function in healthy adults. The most common tests used were serial subtraction tasks (3s and 7s) [29,38,42,51] and Rapid Visual information processing (RVIP) [37,42,45,51]. The cognitive tests are summarised in Table 1.

#### 3.2.2. Lung Function

Three of the cross-sectional, population-based studies [21,22,25] examining the relationship between fruit-derived polyphenols and lung function were used for the meta-analysis. Five cross-sectional studies [21,22,23,24,25], and a single intervention study on human subjects [49], were included in the SR. All data from the cross-sectional studies were extracted from food frequency questionnaires, and intakes of polyphenols correlated to lung function (FEV_1_ and FVC). The single human intervention trial [49] examined the effects of maqui berry extract in 15 asymptomatic smokers with mild cigarette smoking.

### 3.3. Subject Characteristics

#### 3.3.1. Cognition

A total of 409 healthy adult subjects were included in the SR, ranging in ages from 18 to 75 years. The subject sample size ranged from 9 to 40 in studies employing a counterbalanced crossover design [29,35,37,40,41,42,45,46,51], and 30 to 40 in studies using a parallel groups design [38,43,44,53]. The gender of subjects was not stipulated in all studies, but there were a greater number of female subjects (141) than male (99) in those that did [35,37,38,39,42,43,44]. Female subjects therefore represented 59% of the sample.

#### 3.3.2. Lung

A total of 20 788 subjects were included in the SR. All were healthy adults with an age range of 18 to 68 years. Three studies reported gender [21,24,25], with 8513 males and 4551 females represented.

### 3.4. Meta-Analysis Results

#### 3.4.1. Polyphenol Intake and Lung Function

Data from three studies [21,22,25] (comprising 5950 participants) were pooled to examine the effect of polyphenol intakes on lung function in healthy adults. Because the studies included subclasses of polyphenols and fruit (apples and citrus) [25], anthocyanins [21,22], flavan-3-ols [21,22], and flavonoids [22], results from each subclass were considered as independent studies. The summarised adjusted OR shows the overall treatment effect (calculated as the weighted average of the individual ORs). The combined treatment effect of polyphenol intake on lung function is 0.37 (0.15, 0.95, 95% CI, *p* < 0.00001), indicating an improvement in lung function with increased intakes of polyphenols (Figure 2).

#### 3.4.2. Polyphenol Intake and Executive Function

Data from four studies [36,38,41,45] (comprising 150 participants) examining the effects of polyphenol supplementation on executive function were pooled for the meta-analysis. There were no statistically significant effects of fruit derived polyphenol supplementation on cognitive tasks of executive function (SMD = −0.51, 95% CI −1.28, −0.27, *p* < 0.20) (Figure 3). Statistical heterogeneity was high (Tau = 1.53, *p* < 0.00001).

#### 3.4.3. Polyphenol Intake and Attention

Data from six studies [29,35,38,39,45,51] (comprising 171 participants) investigating the effects of fruit-derived polyphenol supplementation on attention tasks were pooled in the meta-analysis. Studies that used two intervention periods (acute and chronic) [38], and/or more than one attention task [38,45,51] were considered s independent studies. The summarised effect of 12 ES shows a small, insignificant effect (SMD = 0.12, 95% CI −0.29, 0.54, *p* < 0.56) of polyphenol supplementation on attention tasks, with significant heterogeneity (Tau = 0.34, *p* = 0.0003) (Figure 4).

## 4. Discussion

The protective effect of dietary polyphenols on numerous chronic diseases considering both disease risk and disease progression are well documented [18,54]. The present systematic review and meta-analysis considered the effects of supplementation with predominantly berry derived products and foods on cognition, and the first to explore lung function in healthy adults. Results suggest that the consumption of various forms of fruit-derived polyphenols exert protective effects on lung function and can improve some aspects of cognition in healthy adults. The meta-analysis of the cross-sectional studies in this review indicates a strong protective effect from various classes of polyphenols, including anthocyanins, on pulmonary function parameters and indicate that diets high in polyphenol rich fruits are associated with better lung function scores (FVC and FEV_1_) in healthy adults. Admittedly the data in the present review were extracted from cross-sectional studies and therefore indicates only that a relationship exists, and not necessarily the cause of such a relationship. 

There is emerging evidence for the role oxidative stress plays in lung inflammation [9]. The pro-inflammatory cytokines: namely TNF-α and IL-1β, and anti-inflammatory IL-10 [55] are associated with lung inflammation and function, and studies using rodent models found an attenuation of both acute and chronic lung inflammation by polyphenol and anthocyanin intake [9,28,29,55]. Mice fed with anthocyanin rich flavonoids have lower nitric oxide (NO) levels, reduced pro-inflammatory cytokines, chemokines, and growth factors 26-28]. Rodent models have shown supplementation with anthocyanin rich berries downregulates inflammatory biomarkers, such as COX-2 [28,54], NF-_Ƙ_B [54], IL-1β, TNF-α [55] and chemokine CCL11 and CCL26 secretion [28,29]. There was only one human intervention trial found for the systematic review. Fifteen asymptomatic smokers showed a reduction in IL-6 following supplementation of anthocyanin rich Maqui twice daily for two weeks [49]. Anthocyanin intake alters innate immune cell trafficking by inhibiting inflammatory biomarkers, indicating a protective role on lung function by reducing oxidative stress. Dietary application of fruit derived polyphenols therefore show promise as a potential novel therapy for lung inflammation, COPD, asthma, or for healthy individuals exercising in a high oxidative stress environment such as air pollution [56]. 

The results from the present meta-analysis on cognition are in alignment with previous reviews, where varying domains of cognition are impacted by polyphenol supplementation [20,57,58]. Eight studies indicated improvements in some aspects of cognition, while six studies showed no effect from fruit-derived polyphenol supplementation. The cognitive tasks that showed improvement were found in both acute and chronic supplementation. Specific brain areas do not perform a single task but are specialised for certain types of tasks, and cognition consists of many domains including memory, working memory, attention, and executive function [11]. The present meta-analysis found a stronger effect on executive function, but not attention tasks. Previous studies have however reported improvements in attention tasks [57] and RVIP [58], with more prolonged effects observed in younger and middle-aged participants. Younger adults also appear to benefit from acute supplementation compared to older adults [57,58]. These are however inferences made from sub-group analyses in studies and no direct comparisons of age-groups using mixed models have been completed. There was a greater proportion of female subjects (59%) than male subjects (41%) in cognition studies in the present systematic review, but how gender effects polyphenol absorption and biological activity, and therefore cognitive effects has not been established in the literature.

Cognitive effects from polyphenol supplementation are likely to be dose dependent, but beyond a point, the bioavailability of anthocyanins decreases with increasing dose [46]. Doses in the present systematic review ranged between 253 mg and 414.2 mg anthocyanins in acute studies, and 143 mg to 1234 mg total polyphenol content. Most of the acute doses were administered 2 h pre-testing as anthocyanin concentrations in the body have been observed to reach peak levels between 1 and 2 h and begin to clear from 6 h [46]. Chronic supplementation doses ranged from 300mg to 600 mg anthocyanins daily, and only one study [44] administered the anthocyanin dose relative to body weight (19.2 mg/g). Administration of chronic doses differed between studies, with daily or twice daily doses (delivering 600 mg/day). Typically, no justification for dosage is given in studies complicating the ability to standardise outcomes or establish an evidence-based reference intake that provides cognitive benefits. 

Anthocyanin or polyphenol dose does not however reflect the actual biological effect achieved from a dose as processing and storage conditions, growing, dietary source, and polyphenol and anthocyanin subclass affect their biological activity [15,59]. Anthocyanin and polyphenol bioavailability is an important issue. Multiple food sources of polyphenols exist, and berries are particularly high in anthocyanins. The therapeutic use of anthocyanins is limited by their low stability and solubility in both organic solvents and aqueous solutions [60], and there are large interindividual variations in absorption, metabolism, and excretion of anthocyanins and polyphenols [61,62], which may explain some of the variance seen across studies with different berry types and variants.

The exact mechanisms of how anthocyanins influence neuronal activity are still not completely understood, yet a number of potential direct and indirect mechanisms of phenolic compounds on brain function have been proposed. It is known that anthocyanins can cross the blood brain barrier [59,63], and animal studies have shown berry anthocyanins can localise in various regions of the brain important for learning and memory [59,64]. The brain is particularly susceptible to excessive RONS due to its high metabolic demand and exposure to environmental toxins [60]. Oxidative stress is implicated in the development of cognitive impairment, and polyphenols and anthocyanins have been shown to modulate and regulate the downstream expression of antioxidant enzymes via nuclear translocation of Nrf2 and the ERK/CREB/BDNF signalling pathway [60,65,66], and increase monoamine neurotransmitter content and inhibit monoamine oxidases in the brain [39]. The anti-oxidant properties of polyphenols may therefore play a role in the observed benefits to cognition.

Endothelial nitric oxide (NO) is considered to be a key molecule related to the proposed benefits, as polyphenols promote NO synthesis, assist flow mediated dilation and neurotransmission [20], and increased cerebral blood flow and regional perfusion in the brain have been found in both human and animal studies following the consumption of polyphenol interventions [20]. Increased neuroplasticity has been reported in the hippocampus following blueberry consumption via stimulation of trophic factors such as brain-derived neurotrophic factor (BDNF) [67].

While the mechanistic data on the effects of fruit derived polyphenols requires follow up, results from the current meta-analysis suggest there is benefit when looking at clinically relevant outcomes, such as cognitive function.

The present systematic review and meta-analyses, while promising, are limited by the number of studies included in each. In the cognition systematic review and meta-analysis, small numbers, as well as the heterogeneity of age groups, cognitive tests and wide variety in anthocyanin and polyphenol content of the interventions used is a limitation. The large variability in the methods for the evaluation of cognition, the cognitive test used (assessing different cognitive domains), and various scoring interpretations, makes it difficult to quantitatively compare studies. The relationship between the intakes of polyphenol rich foods and lung function are based on cross-sectional findings only, and few well controlled studies using healthy, human adults exist. This highlights the paucity of research in both the areas of cognition and lung-function in healthy adults, and further high-quality randomised control trials are warranted in order to bridge the gap from clinical findings to clinical application.

## 5. Conclusions

Lung function in those consuming fruit-derived polyphenols appears to be superior to the non-consumers. With regards to cognitive outcomes, there was significant variation in study type and outcomes measures, and despite pooling results from various studies we did not detect any benefit or decrement. There is a relationship between lung inflammation and cognitive disease and dysfunction, and oxidative stress has been implicated in both. Pulmonary function and respiratory disease have been identified as risk markers for dementia and cognitive impairment. This may be via hypoxaemia or ROS diffusing into the bloodstream due to an overload of the anti-oxidant system in the lungs. The anti-oxidant and anti-inflammatory effects of fruit-derived polyphenols may act as a therapeutic intervention, or as a preventative mechanism in lung inflammation. Further investigation of the relationship between lung function and cognitive impairment is warranted as well as for the mechanistic effects of fruit-derived polyphenols on lung function and cognition.

## Figures and Tables

**Figure 1 nutrients-13-04273-f001:**
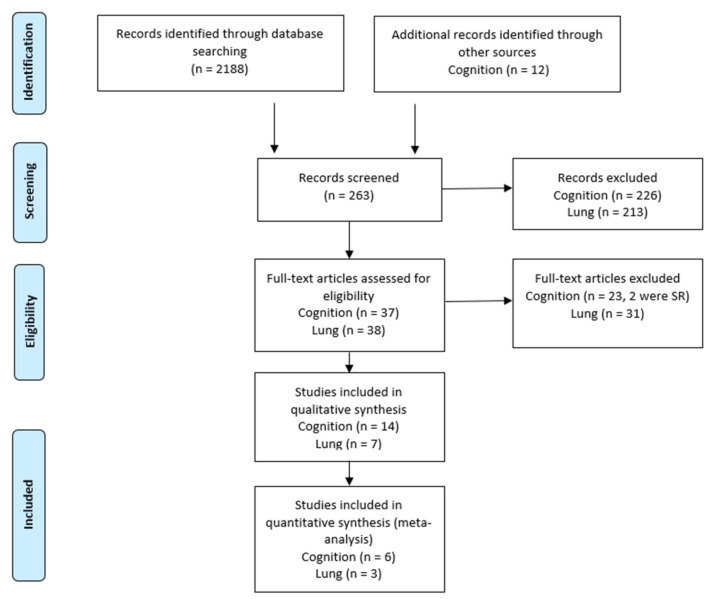
PRISMA Flow diagram of the literature selection process.

**Figure 2 nutrients-13-04273-f002:**
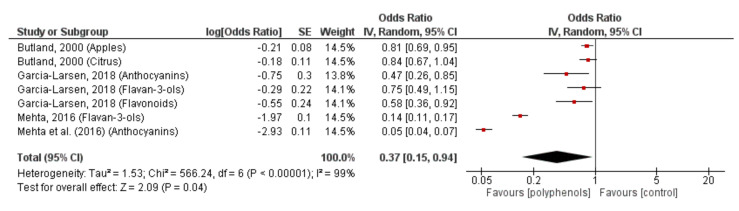
Forest plot of studies investigating the effects of fruit-derived polyphenol intake on lung function (FVC and FEV_1_).

**Figure 3 nutrients-13-04273-f003:**
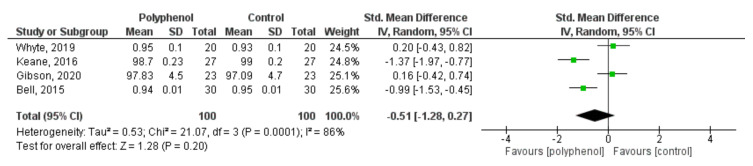
Forest plot of studies investigating the effect of fruit-derived polyphenol supplementation on executive function.

**Figure 4 nutrients-13-04273-f004:**
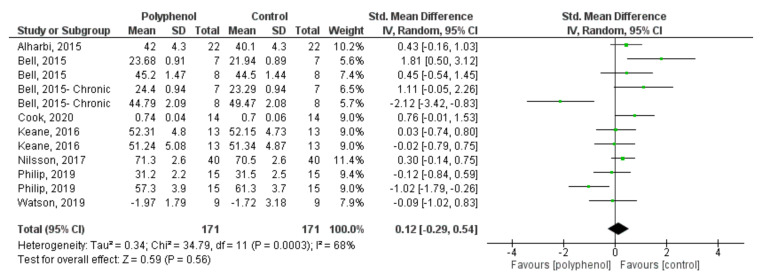
Forest plot of studies investigating the effect of fruit-derived polyphenol supplementation on attention.

**Table 1 nutrients-13-04273-t001:** Summary table of included cognition studies.

Reference	Study Design	Sample Size	Population Details	Study Duration & Outcome Measured	Intervention	Cognitive/Mood Outcomes	Results
Nilsson, et al. (2017) [35]	RCT, crossover Chronic	40	Adults 50–70 years, 30 female, 10 male.	15 weeks: 5 weeks intervention, 5 weeks control, 5 weeks washout between trials. Final dose 9.5–10 h pre.	200 mL Berry mixture (total polyphenol 1234.9 ± 103.2, anthocyanins 414.2 ± 32.8 mg/L, Flavanols 155.9 ± 8.7) or control (matched CHO and pH) consumed 3× per day.	WM, SA	Increased WM at 30 min in BC group. Pronounced learning effect in selective attention.
Whyte, et al. (2019) [36]	Single-blind, RCT, parallel groups Acute	40	Adults 20–30 years	Two consecutive days. Practice on day 1 of full cognitive test battery. Test 1 pre-treatment, then 2, 4 and 6 h following intervention.	400 mL berry ‘smoothie’ (total polyphenol 14.3 g) or PL (matched CHO and Vit C).	MANT, TST, PANAS-NOW.	IncreasedMANT accuracy at 6 h with berry. Faster performance in MANT. TST response time decreased in berry between 2 h and 6 h.
Cook, et al. 2020 [37]	Double-blind, RCT, crossover Chronic + 2 h pretesting.	14	Adults, 12 males, 2 females, 69 ± 4 years	Four lab visits. Physical measures, cognitive assessment and 6 min walk test completed. Visits 1 and 2 familiarisation sessions, maximum of 7-days apart.	300 mg capsules twice daily taken 8 h apart for 6 days PL (cellulose) or NZBC extract (105 mg). Two capsules (600 mg), 2-h prior to arriving at laboratory on day 7. Seven-day washout between trials.	Cambridge neuropsychological test automated battery: RT, SWM, RVIP, PAL.	No difference in RT, SWM, RVIP, PAL.
Bell, et al. (2020) [38]	Double-blind, RCT, parallel groups Acute-on-chronic	60	Adults, aged 18– 30 years, PL = 3 male/27 female, Intervention = 6 male/24 female.	Acute: Tested at baseline, 2, 4, and 6 h post consumption. Chronic: tested at baseline, 6 weeks, and 12 weeks.	Grape seed polyphenol extract (GSPE) (400 mg mix of purified grape seed derived polyphenols) or PL (400 mg maltodextrin) taken daily in capsules.	45 min test in fixed order. AVLT, SSTT, SSTT, MANT, Finger tapping (simple and complex), TST, PANAS-NOW, mental fatigue (Likert scale)	Acute: Faster reaction time with GSPE in switching task and RT in ANT. Better AVLT total recall (*p* < 0.05), switching task accuracy (*p* < 0.1) with PL. Chronic: GSPE improved simple finger tapping score between week 6 and 12 (*p* = 0.005). Limited cognitive benefits from GSPE (acute or chronic).
Watson, et al. (2018) [39]	Double-blind, RCT, crossover Acute	9	Adults, 3 male, 6 female), 23 years	Drink after first cycle of cognitive battery, 45 min post three cycles of cognitive battery completed. 5-day washout in between visits.	96.96 mL Cold pressed blackcurrant juice (BC, 500 mg polyphenols) or PL (sugar, vitamin C, flavour matched).	CogTrack system: RT, DV, CRT. Mood assessment VAS post-completion of cognitive battery, baseline, and end of each repetition of attention tests.	No change in Mood scores. Significant effect for CRT with BC (*p* = 0.028). No change in SRT or DV.
Philip, et al. (2019) [40]	Double-blind, RCT, crossover Acute	30	Adults, 14 male, 16 female, 22 ± 1.7 years.	V1 seven days after V0, and V2 following a 7 ± 2 days washout period. Testing 90 min post drink consumption.	600 mg (2 × 300 mg capsules) of Polyphenol-rich active extract made from grape and wild blueberry (flavan-3-ols, flavanols, anthocyanins = 43.4%) or PL (maltodextrin).	Cognitive demand battery: SSTT, SSTS, RVIP, subjective rating using VAS.	Significant effect of treatment on SSTT: total answers (*p* = 0.001), correct answers (p = < 0.001), errors (*p* = 0.041), % correct answers (*p* = 0.001). No significant effect for RVIP or SSTS. Self-reported cognitive performance significantly higher in PEGB (*p* = 0.033).
Gibson, et al. (2020) [41]	Double-blind, RCT, crossover Chronic + Acute	23	Rugby league players, 28 ± 5 years.	7-day supplementation with treatment or PL, followed by 10-day washout.	250 mL Blackcurrant drink (BD) (Arepa–total polyphenols 1162 mg, anthocyanins 387.5 mg) or PL. Cognitive testing 90 min following ingestion and after fatiguing exercise.	Stroop, subjective mental state measured by mental toughness assessment (MTQ48).	BC - Increase in total Stroop scores (*p* = 0.001), accuracy (*p* = 0.003), and response (*p* = 0.043).
Jackson, et al. (2020) [42]	Double-blind, RCT, crossover Acute	32	Adults, 6 male, 26 female, 22.2 ± 4.2 years.	1 x screening and four test visits. Minimum 7 days between test visits. Treatment drink within 5 min of first CDB. Cognitive assessment conducted 60-, 180-, and 360-min post-dose.	Four treatments: Placebo, Blueberry, Apple, Coffee Berry. Each drink (295 mL) contained one of three polyphenol extracts: Blueberry (BE) (2.49 g blueberry extract, 300 mg blueberry anthocyanins), Apple (AE) (275 mg apple extract, 234 mg flavanols as epicatechin equivalents), or coffee berry (CB) (1.1 g extract, 440 mg chlorogenic acid).	COMPASS Software. Four x 10-min CDB. SSTT, SSTS, RVIP, Stroop, peg and ball, Immediate and delayed word recall, word recognition, picture recognition. Mood assessed by Bond-Lader, mental fatigue and alertness VAS (total test time 40 min per assessment).	No difference in cognitive performance after post-hoc analysis. Alertness significantly higher with AE compared to placebo (*p* = 0.049), as well as lower ratings of mental fatigue (*p* = 0.002) in AE and CB (*p* = 0.003).
Alharbi, et al. (2016) [29]	Double-blind, RCT, crossover Acute	22	Healthy males, 51 ± 6.6 years	Screening visit and each test day separated by 2-week washout. Baseline cognitive battery (CB) followed by consumption of either FR or PL. Cognitive testing at 2 h and 6 h post drink consumption.	240 mL flavonoid rich orange juice (FR, 272 mg flavonoids) or PL matched for volume, taste, appearance, energy, and glucose.	Cognitive battery: Immediate word recall, simple and complex finger tapping, DSST, CPT, SSTS, PANAS, contrast sensitivity, delayed word recall. Total test time: 45 min	Significant increase in simple finger tapping following FR 2 h post (*p* < 0.01) and 6 h post (*p* < 0.05). Significant drink x time observed for CPT accuracy (*p* < 0.05) at 6 h. Subjective alertness ratings significantly higher following FR (*p* = 0.05).
Khalid, et al. (2017). [40]	Double-blind, RCT, crossover Acute	21	Young adults, 20.14 ± 1.01 years.	Three testing sessions separated by minimum 3-day washout period. Screening day all received placebo. Two subsequent testing either PL or WBB. Each visit, baseline mood measures conducted, and 2 h post drink consumption.	250 mL Flavonoid-rich wild blueberry (WBB, 253 mg anthocyanins) or PL (matched for Vit C and sugars).	PANAS-NOW	Increase in positive Affect (PA) after WBB consumption (*p* = 0.001).
Bowtell, et al. (2017) [43]	Double-blind, RCT, parallel groups Chronic	26	Adults, 13 female, 13 males, 65–69 years.	Baseline cognitive function testing, with follow up 12 weeks.	30 mL blueberry concentrate (BBC, 387 mg anthocyanidins) or PL (synthetic apple and blackcurrant cordial) consumed once per day for 12 weeks.	CogState Ltd.: RT, Groton maze and timed chase and learning test, International shopping list task with delayed recall, 1-back, 2-back memory tasks	No difference between BBC and PL.
Miller, et al. (2018) [44]	Double-blind, RCT, parallel groups Chronic + Acute	37	Adults, 13 male, 24 female, 60– 75 years	First dose of supplement taken after testing. On day 45 (visit 3), supplement consumed, and cognitive testing performed (except for CVLT-II and TMT). Day 90 (study visit 4), final supplement dose consumed, and cognitive testing completed.	12 g lyophilized Tifblue blueberry, twice daily (24 g/day, approx. 36 mg/g total phenolics, 19.2 mg/g anthocyanins), or PL for 90 days.	TST, TMT, CVLT-II, DST, vMWM, ANT, POMS.	Decrease in task switch stimuli errors across trials with blueberry (*p* = 0.044).Blueberry fewer repetition errors in CVLT on visit 4 than visit 2 (*p* = 0.032). No difference in any other cognitive measures.
Keane, et al. (2016) [45]	Double-blind, RCT, crossover Acute	27	Healthy adults, 50 ± 6 years.	Three separate visits to lab. Baseline cognitive testing then drink (BC or PL) consumed. Cognitive assessments at 1, 2, 3, and 5 h post consumption.	60 mL dose Montmorency cherry (MC, 68 mg cyanidin-3-glucoside/L, 160.75 mean gallic acid equiv/L) concentrate or PL (fruit flavoured cordial).	COMPASS software: DVT, RVIP, Stroop x 2. VAS for alertness, fatigue, and level of difficulty.	No difference in any cognitive or mood scores.
Igwe, et al. (2020) [46]	Double-blind, RCT, crossover Chronic	28	Healthy adults, (55 + years)	Baseline testing followed by 8 weeks of dink consumption, 4-week washout. Eight-week consumption of alternate beverage. Total 20 weeks. Data collection at baseline, 4 weeks, 8 weeks, 16 weeks and 20 weeks.	200 mL Queen Garnet Plum (QGP) nectar (anthocyanin 7.4–10.6 mg C3G* equiv) or PL per day for 8 weeks, then alternate drink for 8 weeks.	Cognitive tasks: RVALT, verbal fluency task, digit-span backwards task, Stroop, counting span.	No difference from baseline or between drinks.

MANT—Modified Attention Network Task, TST—Task Switch Task, PANAS—Positive and Negative Affect Scale, RT—reaction time, SWM—spatial working memory, RVIP—Rapid Visual Information Processing, PAL—Paired Associates Learning, AVLT—Auditory Visual Learning Task, SSTT—Serial Subtraction Task 3′s, SSTS—Serial Subtraction Task 7′s, CRT—Choice Reaction Time, DV—Digit Vigilance, CDB—Cognitive Demand Battery, POMS—Profile of Mood States, DSST—Digit symbol substitution Test, CPT—Continuous Performance Task, TMT—Trail Making Test, CVLT-II—California Verbal Learning Memory Test, DST—Digit Span Task, vMWM—Morris Walter Maze, RVALT—Rey Auditory Verbal Learning test * C3G—Cyanidin-3-glucoside.

**Table 2 nutrients-13-04273-t002:** The effects of fruit-derived polyphenol consumption on lung function in healthy adults.

Reference	Study Design	Sample Size	Population Details	Study Duration & Outcomes Measured	Intervention/Method	Outcomes	Results
Mehta et al. (2016) [21]	Longitudinal analysis.	839	Males, 65–68 years	12 years. FFQ and lung function assessed every 3–5 years. Mean follow-up time: 7.4 y.	Self-administered FFQ. Flavonoid subclasses extracted. Intakes (mg/d) of anthocyanins, flavanones, flavan-3-ols, flavanols, flavones, and polymers. FEV_1_ (mL), FVC (mL).	FEV_1_ (mL), FVC (mL).	Higher daily anthocyanin intake (25.3 mg/d) slower rate of FEV_1_ decline relative to low anthocyanin intakes (1.3 mg/d). FVC had slower rate of decline (37.3 mL/y) with high anthocyanin intake (25.3 mg/d) vs. low intakes (1.3 mg/d). Blueberry intake (≥ 2 servings/week) associated with slower rate of lung function decline in FEV_1_ (22.5 mL/y) and FVC (37.6 mL/y).
Vergara, et al. (2015). [49]	Exploratory, uncontrolled study.	15	Asymptomatic smokers, with low F&V intake (<200 g/d) and moderate cigarette smoking (8 cigarettes/day). Age 26.4 years.	2-weeks. Expired breath condensate (EBC) collected before and after administration.	2 g maqui extract ingested twice daily for 2 weeks. Total polyphenol content of extract: 5.18 ± 2.00 g Gallic Acid Equivalents (GAE)/100g	Hydrogen Peroxide (H_2_O_2_) and IL-6 in EBC.	Significant decrease (57.3%) of H_2_O_2_ concentration (*p* = 0.0002), and increase (30.8%) in IL-6 concentrations (*p* = 0.0032) observed in 92.8% and 80% of smokers respectively.
Garcia-Larsen, et al. (2018). [22]	Multi-national, population-based, cross-sectional	2599	Adults, 15–75 years.	2008–2009.	GA^2^LEN FFQ to assess dietary intakes across Europe. Six main flavonoid subclasses measured: Anthocyanins, flavanones, flavan-3-ols, flavanols, flavones, and polymers. Total flavonoid was the sum of all six subclasses.	FEV_1_/FVC ratio, FVC (mL).	53% lower risk of FVC < lower limit of normal (LLN) in adults with highest anthocyanin and pro-anthocyanidins intake (*p* = 0.04), 42% lower risk of FVC < LLN in those in highest quintile of flavonoid intake (*p* = 0.07).
Tabak, et al. (2001). [23]	Cross-sectional	13651	Adults, 20–59 years	1994–1997	Semiquantitative FFQ questionnaire. Flavonoid intake calculated using specific food composition tables.	COPD symptoms questionnaire, FEV_1_	Total intake of catechins (not from tea), flavanols and flavones positively associated with FEV_1_, and inversely associated with COPD (adjustment for smoking reduced the observations).
Pounis, et al. (2018). [24]	Cross-sectional	9659, (4551 women, 5108 men)	Adults, ≥ 35 years	March 2005–April 2010	FEV_1_, FVC (mL), FEV_1_/FVC ratio. Pulmonary symptoms determined by questionnaire at recruitment. EPIC-FFQ to determine nutritional intakes in the past year.	Total intakes: flavanols, flavones, flavanones, anthocyanidins, isoflavones, lignans, all as mg/day. Dietary index PAC score, FEV_1_, FVC (mL), FEV_1_/FVC ratio.	Increase in PAC score associated with increase in all pulmonary function parameters in women *p* < 0.05), and FEV_1_ % predicted and FVC % predicted in men (*p* < 0.05).
Garcia-Larsen, et al. (2015). [50]	Cross-sectional	1187	Adults, 22–28 years	January 2001– April 2003	Semi-quantitative FFQ of 65 food items. Total F&V intake, nutrient estimates, and flavonoid content of foods assessed: flavanols, flavones and catechins.	FEV_1_, FVC (mL), FEV_1_/FVC ratio	Higher FVC observed in highest vs lowest quintile of flavanol (80 mL higher), and catechin intake (70 mL). Significant association between FVC and total intake of fruits between highest and lowest quintiles (*p =* 0.006); catechins after adjustment (*p* = 0.02), but not vegetables.
Butland, et al. (200). [25]	Cross-sectional	2512	Adult males, 45–49 years	1979–1983	Phase I data collection and self-administered semi-quantitative FFQ, as well as lung function tests. Phase II after 5 years, dietary data collected through slightly modified FFQ and respiratory function tests.	FEV_1_, FVC (mL)	Lung function positively associated with citrus fruit when adjusted for age and height but not after full adjustment. Lung function higher for those eating 5 or more apples a week compared to non-consumers.

## Data Availability

Data is contained within the article or Appendix A.

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
