# Peer review of "The Effects of Fruit-Derived Polyphenols on Cognition and Lung Function in Healthy Adults: A Systematic Review and Meta-Analysis"

_nutrients, 2021, doi:10.3390/nu13124273_

Round 1
Reviewer 1 Report
Importantly, I should state that I am not familiar with the processes for conducting systematic reviews and meta-analyses and so my below comments pertain only to the content of the paper, not whether the procedures for performing the review are correct. That is why I have selected ‘not applicable’ on the 3 design/results points on my reviewer report form.
For me, the rationale to combine cognitive and lung research in this 1 paper is not present; is there some link I’m missing? It makes reading sections like the discussion really difficult as what outcome, cognitive or lung (or both)? are you conjecturing to be benefited by increased cerebral blood flow?, as an example.
Initially, the authors need to be clearer in what the mean by fruit-derived polyphenols as the review excludes many key papers that I would consider are missing, based on my understanding of a ‘fruit-derived polyphenol’. For example, resveratrol is derived from grapes and so my own papers (Wightman et al. and Kennedy et al., from 2010 onwards) are missing from this review as well as those from Howe and Wong in Australia, just off the top of my head. It’s not until the first sentence of the discussion that you note the exclusion of resveratrol, in favour of berry polyphenols, but the rationale for this distinction needs to be clearer from the abstract onwards.
Specific comments:
- Line 11 of abstract, spell out ‘MA’
- If ‘MA’ abbreviates meta-analysis throughout I would revise as I think abbreviating, unless absolutely necessary, really breaks up the ability to read fluently
- Line 16 of abstract, spell out PRISMA
- Line 54 of introduction should read …..”interventions in younger……”
- Line 57 of introduction states that there are few trials reporting cognitive effects in young people but this review (reference number 21) was published almost 7 years ago and there have been many studies in this area since then.
- Line 62 of the introduction requires citations after …”cancer types.”
- Line 98 of the introduction should read ….”needed to include…”
- Line 99 of the introduction needs to spell out FVC and FEV
- Line 106-107 in section 2.2. states that work was included if it was published in the period between August 2020- October 2021 but, of course, the articles included preceded this time.
- Could the references in the table (i.e. Jackson et al. (2020) also include their numerical value in the reference list (i.e. Jackson et al. (2020) 32)?
- Why does the legend underneath table 2 on lung function (lines 225-229) discuss cognitive function? Should this be under table 1?
- The first paragraph of the introduction is too long and too vague; you talk about doses and tasks but not what these are.
- Overall, I find the discussion very weak. It is missing a clear structure and specific information; e.g. discussion of doses/levels/types of polyphenol/type of participant etc etc, is missing. There really isn’t any critical discussion around these key themes. I think that this section just needs to be completely revised and, if keeping the combination of cognition and lung research in this 1 paper, then I’d suggest very clearly separating those effects and conjectured mechanisms between lung and brain.
Author Response
Thank you for the time you took to review and respond to our recent article submission. It is greatly appreciated. Your feedback was valuable and I hope we have satisfactorily addressed them in the attached responses, and with the changes made to the submission.
I have not included line references in the attached due to the large number of changes that have now been made to the article.
Thank you again.

Reviewer 2 Report
In this study, a comprehensive systematic review was conducted to evaluate the effects of fruit-derived polyphenol intake on cognitive and pulmonary function in healthy subjects. This review is well written. To the reviewer's knowledge, there has been no systematic review on this topic in the past, and it will provide important knowledge and useful information for this research area. However, there are some comments that could elevate the quality of this paper. Reviewer added recommendations and suggestions regarding this.
Comment 1
This review focuses primarily on cognitive function and pulmonary function in healthy individuals. Are there any physiological functions other than cognitive and pulmonary function that have been elucidated by polyphenols, such as risk of lifestyle-related diseases and life expectancy in healthy adults? Also, are there any reported synergies between cognitive function or lung function and other physiological effects? It would be more interesting for the reader to add these explanations in the first half of the article.
Comment 2
Reviewer recommended to create a visible diagram of the classification of the subclasses of polyphenols and anthocyanins in fruits (especially berries, as described in this review).
Comment 3
Authors mentioned in this review as their significance of polyphenols on lung is already known, which seems a bit hard to understand. Therefore, reviewer recommend to more emphasize the minority and importance/significance of investigating the effects of anthocyanin on (1) lung and (2) cognition in healthy individuals.
Comment 4
Reviewer concern about the subjects’ characteristics used for this analysis. Authors mentioned about their age but not about their sex even though some investigations the number of male/ female is considerably biased. How was the ratio of male and female used in this analysis: is the sex a considerable factor?
Author Response
Thank you for the time you took to review our recent submission and the feedback you provided. It is much appreciated. Please see the attached for responses to your comments and feedback.
Thanks again.

Round 2
Reviewer 1 Report
This revised manuscript clearly shows an exhaustive overhaul in such a short space of time and the authors should be commended for this. In large part, I am happy with the revisions based on my previous comments but I would still like to see mention in the introduction of why certain classes of fruit-derived polyphenols have been omitted from this meta-analysis. Especially as the discussion now states that a critical limiting factor was the low number of studies in this area and, it could be argued, that this is because you have omitted a big body of this work; especially in the resveratrol field. If there is a reason for omitting this work, which I presume there is, then why not insert it?
Author Response
Thank you for your expedient reply, and also your comment acknowledging the work required for the re-write. You have raised a valid point regarding resveratrol, however we chose to define fruit derived polyphenols as fruit extracts, products or whole food interventions. Resveratrol as a single nutrient supplement was excluded, but grape products and extracts have been included.
I have amended the introduction (lines 84 and 85) using the above definition and hope that this in some way addresses your comments.
Thank you again for the response and review.